# Emerging Market Default Risk Charge Model

Angelo D. Joseph 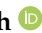

School of Business Leadership, University of South Africa, Midrand, Johannesburg 2006, South Africa; angelo@quantanalyst.co.za

**Abstract:** In a default event, several obligors simultaneously experience financial difficulty in servicing their debt to the point where the entire market can experience a sudden yet significant jump to a credit default. To help protect lenders against a jump-to-default event, regulators require banks to hold capital equivalent to the default risk charge as a buffer against the losses they may incur. The Basel regulatory committee has articulated and set default risk modelling guidelines to improve comparability amongst banks and enable a consistent bank-wide default risk charge estimation. Emerging markets are unique because they usually have illiquid markets and sparse data. Thus, implementing an emerging market default risk model and, at the same time, complying with the regulatory guidelines can be non-trivial. This research presents a framework for modelling the default risk charge in emerging markets in line with the regulatory requirements. The default correlation model inputs are derived and empirically calibrated using emerging market data. The paper ends with some considerations that regulators, supervisors, and banks can use to get financial institutions to adopt an emerging markets default risk charge model.

**Keywords:** emerging markets; default risk charge; correlation modelling; framework

## 1. Introduction

In the aftermath of the 2008 financial crisis, the Basel Committee for Banking Supervision (BCBS 2009a) revised the calculation of the capital banks are required to hold as a buffer against losses that they may incur in the event of a credit market default. The main reason for the revision was to account for changes in obligor credit quality and default correlations[1]. Pre-2008, the banks generally relied on the Basel II Value-at-Risk (VaR) measure to determine regulatory capital (Jackson et al. 1997). However, the credit market volatility in the financial crisis quickly proved that prudent risk management of all the risk types requires that financial institutions augment their traditional risk measures (BCBS 2009b). To get an overall perspective of all the market risks, institutions incorporated stress events by stressing the VaR and supplementing the stress VaR with an incremental credit risk charge (IRC) to account for the credit quality or default risk[2].

One benefit of using the IRC model is that the regulatory authorities published a prescribed modelling methodology for determining this credit charge (BCBS 2009b). However, the regulatory-prescribed IRC model is an oversimplified approach in that it is limited to capturing the default risk by modelling the economy using a specified one-factor[3] model. Furthermore, the IRC model is designed to capture the complex credit rating migrations, and defaults for general interest rate-sensitive instruments and optionally includes equity (Xiao 2009). Moreover, under the IRC model, it is not necessary to determine a charge for instruments that are equity sensitive, even though the typical losses-given default on equity is 100% (Rodrigues and Maialy 2018).

In the fundamental review of the trading book (FRTB), the regulator noted that the IRC with its more complex credit state migrations was the main source of the large variability and incomparability of capital charges across banks (BCBS 2014). The IRC variability was found to be problematic in that it made it difficult for the regulator or supervisor to compare

and assign confidence to the capital charges determined by each participating bank (Laurent et al. 2016). There was now a definite need for smarter, non-complex and tougher responses to the global financial crisis (Prorokowski and Prorokowski and Prorokowski 2014). As an alternative, the regulator proposed a more prescriptive IRC, a default-only simulation model with at-least two factors, that excludes the complex migration risks[4]. This more prescriptive charge for credit default risk is called the default risk charge (DRC). The DRC is defined as a VaR type measure aimed at capturing default correlations and jump-to-default risk for both interest rate sensitive and equity instruments (BCBS 2016).

Even though the regulator somewhat simplified the default risk model to exclude the complex credit migrations, implementing the model (especially in emerging markets) and complying with the prescribed regulations can be non-trivial (see e.g., the default risk models in Wilkens and Predescu 2016; Laurent et al. 2016; Slime 2022). One particular difficulty arises from the fact that the DRC, as prescribed by the regulator, was possibly derived assuming developed, liquid and more perfect credit markets. On the other hand, emerging markets are unique in that they are generally illiquid, segmented and lack historical financial databases (Al Janabi 2006). This paper aims to present a simulation-based DRC model in an emerging market setting that is well in line with the regulatory requirements. To enhance the quality of the default simulations, the default correlation model inputs are derived and empirically calibrated using emerging market data. Experimental tests are then performed to establish the soundness of the derived emerging market DRC model. The paper ends with some guidelines that regulators, and bank supervisors can adopt for sound emerging market default risk modelling.

## 2. Default Charge Estimators

The DRC measure aims to estimate the capital that should be held as a buffer against losses that a financial institution can incur in the event of a sudden jump to default or stress credit market event. The DRC model is designed to project default losses over a one-year capital horizon at a 99.9% confidence level (BCBS 2019). The DRC is unique in that it is not a general credit spread risk measure but is more concerned with the capital at risk in the event of a jump to default[5]. To effectively capture the jump to default risk, regulators recommend that banks consider using one of two estimators: the standardised or the internal models approach DRC.

### 2.1. Standardised Approach

The standardised approach DRC is based on pre-defined default model parameter approximations (BCBS 2014). One particular approximation in the standardised DRC is the prescribed risk weights. The risk weights are a proxy to the credit default probability and are assumed to be constant over all issuers in a particular credit rating band. See Figure 1.

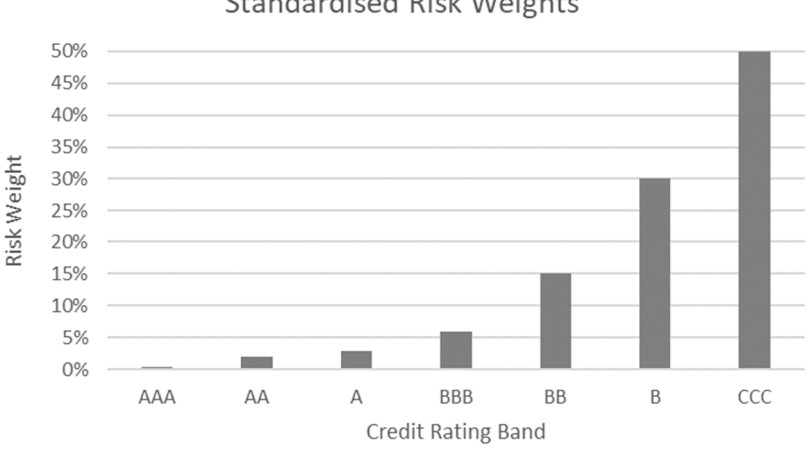

**Figure 1.** Standardised approach risk weights per credit risk band.

Moreover, this means that under the standardised DRC, an issuer with a particular credit rating and a positive outlook (e.g., AA+) has the same risk weight as an issuer that has no outlook (e.g., AA) or a negative outlook (e.g., AA−). This means that the standardised DRC assumes the same constant default probability for all issuers within a particular rating band (e.g., AA+, AA, and AA− rated issuers). It is well known that the credit outlook of an issuer impacts the credit risk expectations and, therefore, the default probability (see e.g., Hull 2015). The use of the constant risk weights per rating band can therefore render the standardised default risk charge too conservative for (no and positive outlook rated) issuers that are not necessarily as aggressively priced (as negative outlook rated credits) on a probability of default basis. Even with the latest revised standardised approach that somewhat incorporates the credit outlook of the issuer (Feridun and Ozün 2020), due to the constant approximation nature of the risk weights, the possible over-conservatism with the standardised default charge may unfortunately still be present.

The standardised approach, however, does have advantages. Firstly, the standardised approach's approximations could be helpful in the estimation of the default capital charge in markets where default probability data is generally sparse. Illiquid markets and infrequent credit default events usually characterise emerging credit markets (Clift 2020). Therefore, emerging market financial institutions usually require much data to estimate a statistically significant probability of default (PD)—not to mention the remedies and science required to be applied to the data to calibrate a statistically sound emerging market loss-given-default (LGD) model. The standardised DRC approach does not pose data limitation risks in estimating the default charge due to the usage of the regulatory prescribed model parameters (like the prescribed risk weights seen in Figure 1).

The use of the standardised DRC can, unfortunately, be more biased to overstating the capital charge because of the regulator-prescribed or predetermined "one-shoe-fits-all" (risk weights or PDs and LGDs) model parameter estimates (Laurent et al. 2016). The misstated capital charge is possibly worse for emerging markets, with the extra margin of conservatism coming from approximations or fudges employed to manage the data of poor quality. In this light, many financial institutions consider[6] the internal models approach DRC.

### 2.2. Internal Models Approach

The internal models approach is a radical change from the standardised approach (Jackson et al. 1997). The rationale and technical specification underpinning the "radical" internal models approach DRC is given below.

### 2.2.1. DRC Rationale

The monetary losses experienced by banks due to the 2008–2009 subprime credit crisis, drove the realisation that using only the VaR metric for raising capital could be inadequate (BCBS 2009b). The sub-prime credit crisis established that even after scaling up to 10 days, the traditional 1-day $VaR_{99\%}$ is inadequate for capital purposes for two reasons. Firstly, by choosing a 99%-tile VaR, the risk manager assumes that the bank will not experience losses more frequently than two to three times in a trading year (1-2/252 = 99.21% to 1-3/252 = 98.80%) (Hull 2015). Unfortunately, the 2008 credit crisis showed banks that the 99% tile VaR assumption is not conservative, in that capital can be understated if it is based only on the 99%-tile VaR. Secondly, the scaling of the one-day VaR up to 10 days may only partially reflect the potential for large cumulative price movements over several weeks or months. Furthermore, scaling the 1-day VaR is technically problematic because the return distribution is generally different over various return time frames. In fact, based on multi-asset portfolio data, Jackson et al. (1997) argue that the ten-day returns serve to underline the statistical problems involved in attempting to deduce ten-day volatilities directly from estimates of one-day volatility. The 10-day $VaR_{99\%}$ may therefore not fully incorporate the liquidity of the trading book, as seen in the understatement of the losses due to (amongst other problems) liquidity in 2008.

When the regulator realised the shortcomings of the 10-day VaR$_{99\%}$, they introduced the incremental default risk charge and, more recently, the default-only risk charge[7]. In response to the increasing amount of exposure in banks' trading books to credit-risk-related and often illiquid products whose risk is not entirely reflected in the traditional VaR, a credit-based VaR-type metric was introduced (BCBS 2009b). The internal model-based default risk charge, designed to be a higher 99.9%-tile VaR, simulated over a 1-year forward time horizon (BCBS 2016), was introduced in order for it to effectively supplement the traditional VaR.

### 2.2.2. DRC Specifications

The internal models approach (IMA) DRC is prescribed to be a two-factor VaR type default simulation at the 99.9% confidence level (BCBS 2016). In the IMA DRC, the choice of the two factors is not regulatory prescribed, and the model is recommended to simulate the possible default states using the banks' internally rated PDs (BCBS 2019). (This is probably why regulators coined this DRC approach an internal models approach). Due to the high 99.9% percentile nature (relative to the traditional 95%-tile or 99%-tile general VaR) of the IMA DRC, many simulation paths are required for reasonable precision (Slime 2018). The IMA DRC can therefore be a computationally intensive default simulation requiring hundreds of thousands or millions of simulation paths for reasonably good precision.

The IMA DRC does have benefits. Its simulation nature makes it a much more realistic and sensible approach to estimating the default risk capital charge. The realistic and sensible nature of the internal model approach DRC make it a plausible candidate for estimating the capital charge in emerging markets—where the data quality poses an extra risk dimension of misstating the DRC. An emerging market credit model estimation is generally highly impacted by the data quality and can therefore be subject to many data assumptions and approximations. The internal models approach is less prone to approximations (compared to the standardised) and allow the modeller the transparency to monitor and manage the model assumptions more effectively (Ferreiro 2016). Thus, the internal models' approach to DRC can be practically argued to be a better choice in emerging markets.

From the academic perspective, Laurent et al. (2016), Wilkens and Predescu (2016) and Slime (2022) argue that the IMA DRC is more palatable to financial institutions since it generally results in lower capital charges than that of the standardised approach.

Unfortunately, like financial risk management works, the lowered capital charge with the IMA DRC does not come for free. Similar to using the cost-saving Comprehensive Risk Measure (CRM) (Prorokowski and Prorokowski and Prorokowski 2014), a bank must get regulatory approval before using the internal-based DRC. The approval will, in all likelihood, be dependent, amongst other requirements, on how well the implementation of the model complies with the regulatory guidelines (See Figure 2).

An essential IMA DRC input requirement worth discussing is the probability of default (or the PD). Banks are recommended to either use their internal-ratings-based approach or an objective, real-world PD estimate. The PD is essential because it defines the jump-to-default threshold in the default simulation (to be discussed in Section 3.2) and is, therefore, one of the main drivers of the capital charge. For this reason, the regulator gives first preference to the use of the banks' own (regulatory-approved) PDs in estimating the default risk charge. Given this heavy capital dependency, an issuer's default probability (and loss given default, LGD) should, in theory, be based on the bank's own internal estimates. Since using internal parameters could open up the banks to manipulate the default charges, to retain some objectivity, it is a good idea for the regulator to require banks to first get regulatory approval of the IMA DRC model before its effective use.

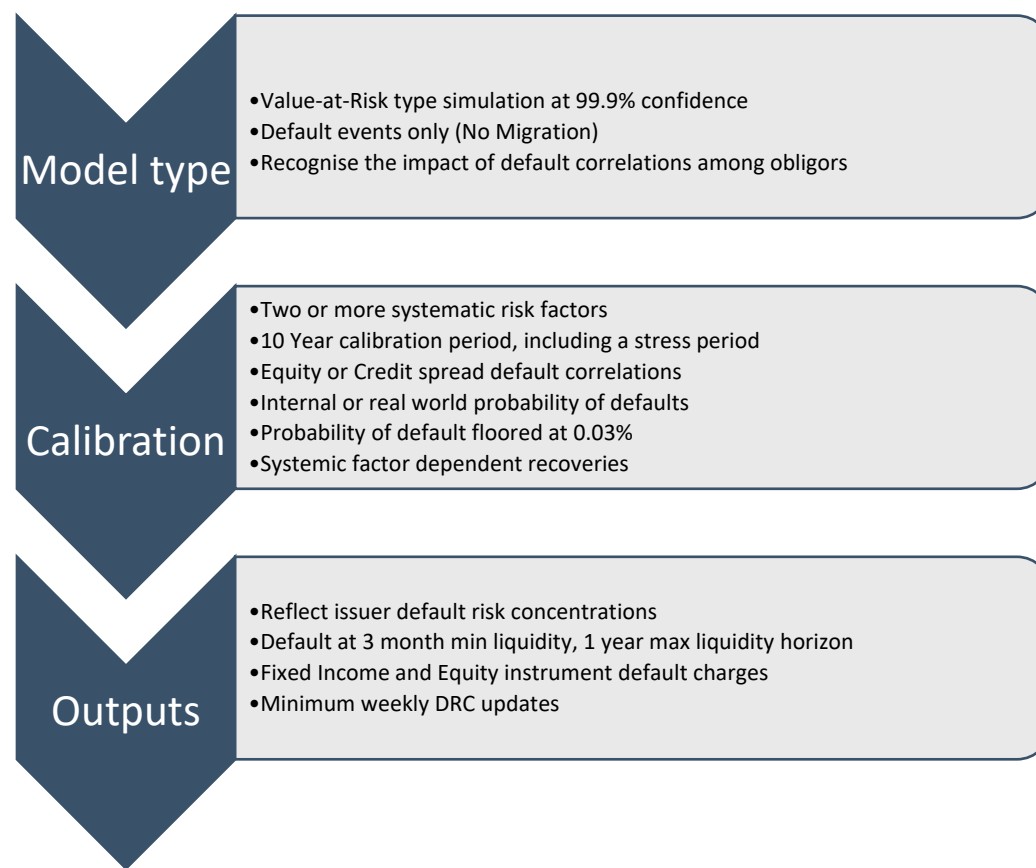

**Figure 2.** Basic attributes of the internal models approach DRC model. Adopted from the Basel Committee on Banking Supervision regulatory guidelines (BCBS 2016).

Default Correlation

Undoubtedly, correlations are among the most challenging and least understood facets of risk management (Prorokowski and Prorokowski and Prorokowski 2014). The aspect of the default correlations in the IMA DRC (amongst other pertinent model input requirements -see the non-exhaustive list in Figure 2) could make the model appear non-trivial to implement. Moreover, a complex modelling set-up may be considered expensive and therefore lower a financial institution's implementation appeal of the IMA DRC.

Interestingly, the IMA DRC could present a viable trade-off between capital savings and implementation cost. Capital savings linked to DRC accuracy are important. In fact, the Basel IV reform implementation was initially due on 1 January 2022, but due to the adverse impacts seen from the coronavirus disease 2019 (COVID-19) on the global banking system, it was postponed to 1 January 2023 (Feridun and Ozün 2020). Possibly overestimating capital charges due to forcing banks to conform to a particular default charge model is definitely not on the agenda of the BCBS members. Thus there is a strong case for potentially using the IMA DRC from an accuracy and capital-saving perspective, especially when markets are less volatile and capital at risk is generally expected to be lower.

Even though the IMA DRC appear non-trivial due to its effective usage demanding an understanding of default correlations, one cannot disregard the fact that the IMA DRC provides a viable framework in which the credit risk of the trading book positions, contained in credit quality linked assets, can be dynamically measured (Laurent et al. 2016; Slime 2018). The default correlation in the IMA DRC has even wider comparative benefits. Moreover, for credit-bearing portfolios where there is zero default correlation between issuers[8], the IMA DRC is smaller than the standardised-based DRC regardless of the portfolio size (Slime 2022). Moreover, the addition of uncorrelated securities amplifies

the internal models-based DRC benefits against the standardised default charge, which sums the weighted jump-to-default for all issuers (Wilkens and Predescu 2016).

In this paper, we focus only on the IMA DRC model in order to provide a framework that complies from a regulatory standpoint to estimating the credit risk capital charge in emerging markets.

## 3. Emerging Market

As a result of the significant macroeconomic differences between emerging and developed markets, emerging markets have unique, relevant local data that do not exist or are atypical (Al Janabi 2006; Cabrera 2022). Some emerging markets have no companies in specific sectors, such as Brazil and Russia, which have no locally quoted biotechnology companies (Pereiro 2010). Others have highly biased markets, such as Argentina, where the oil industry accounts for more than 40% of the market capitalization (Cabrera 2022), and Slovenia, where the pharmaceutical industry represents more than 1/3 of the market (Pereiro 2010). Unfortunately, the emerging market sector beta generally represents the market and not necessarily the sector.

In this paper, the South African market is chosen as the emerging market for two reasons. Firstly, the South African market is highly exposed to the mining industry (Wikipedia 2022), thus representative of emerging markets that are biased toward a particular sector. Secondly, South Africa has been ranked among the world's fastest growing and emerging market economies for years (Economic Times 2023). Therefore, the South African market is a sound sample representative of the emerging market population.

### 3.1. Default Simulation

When it comes to emerging markets and their underlying data nuances, effectively measuring and managing the market risks requires, amongst other prerequisites, transparency. Using a simulation DRC approach is one way of improving transparency amongst dependencies and individual factor risks. To appreciate the transparency in simulating the IMA DRC, it is imperative that the primary simulation mechanism underlying the default risk estimator be considered. Moreover, the simulation mechanism is essential to contextualize the multi-factor economic model and the default correlations in estimating the default charge. This section outlines the primary simulation mechanism underlying the IMA default risk charge estimation.

### 3.2. Simulation Process

The process of simulating the DRC is relatively straightforward. Firstly, assume an obligor with default definitions given by the PD and LGD. Generate a matrix of random drawings from a standard uniform distribution. Suppose each obligor's random trajectory is governed by a multi-factor systemic economic model (to be discussed in Section 4) that defines a default factor-based correlation matrix. Then using the random draws and the correlation matrix, simulate a default index for each obligor. For each obligor, if the 1-year simulated default index is lower than a specific default barrier (defined using the PD), register that obligor as ending up in default. If the obligor does not end up in default, go to the next obligor. For all the obligors that ended up in the default state, determine the Jump-to-Default (JtD) monetary loss. The JtD is the expected capital loss and is determined as the monetary exposure at default incorporating the LGD. The regulations specify that the JtD is the nett loss of the principal amount at default after accounting for the mark-to-market loss (BCBS 2016; Ferreiro 2016). This process is repeated for many simulations runs such that a default loss distribution is formed. From the loss distribution, the 99.9%-tile value is calculated and recorded as the DRC. For ease of understanding, a mathematical summary of the default simulation process is given in Appendix A, while a flowchart of the simulation process is given in Figure 3.

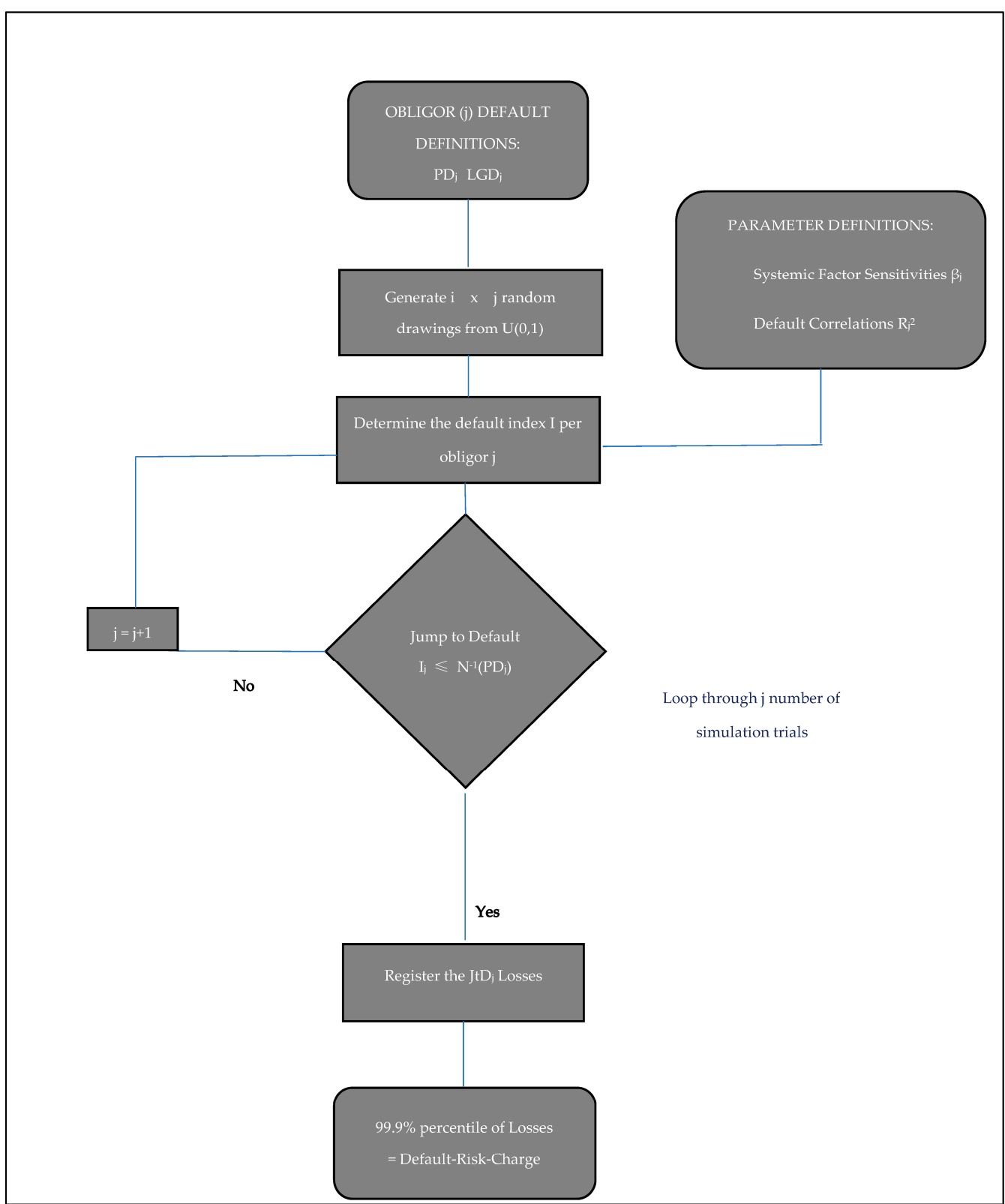

**Figure 3.** Schematic presentation of the DRC simulation process.

The flow chart in Figure 3 reveals that the simulation process underlying the DRC is simple and transparent. However, it is interesting to note that the default charge (and surely its accuracy and precision) by simulation depends on the choice of systemic factors and their correlations.

## 4. Modelling

The simulation process outlined in Figure 3 illustrated that the quality of the DRC simulation is dependent on the systemic factors, and therefore the model used to estimate the systemic factors and their correlations. This section is devoted to the important default risk systemic factor models.

### 4.1. Default Risk Factor Modelling

The regulator prescribes the DRC to be a simulation-based default model with at least two systematic risk factors (BCBS 2019). Unfortunately, the academic literature on DRC factor modelling is scant. One fascinating default risk simulation model that is based on the eigen-decomposition of the correlation matrix was published by Laurent et al. (2016). The eigen-decomposition of the systemic factor correlations is used to select the two factors that best describe the dynamics of the systemic factors over the 10-year calibration period. The eigen-decomposition is central to analyses designed to find the principal components that drive the factor correlation matrix. It is, therefore, typical to do a principal component analysis to aid in selecting the type and best set of systemic factors (Slime 2018). The drawback of an eigen-decomposition approach is that the eigenvectors are not easily identifiable to economic variables. This way, an eigen-decomposition of the systemic correlations could provide opaque systemic factors. The unidentifiable factors under the eigen-decomposition may complicate the issuer default risk concentration analysis.

A default risk simulation model that enables a risk manager to identify (at least) two factors and relatively easily relate them to economic variables was published by Wilkens and Predescu (2016). In fact, the Wilkens and Predescu (2016) model is based on economic risk factors in a regression setting, making their model much more transparent and traceable for effective factor risk management. Will the fact that the Wilkens and Predescu (2016) model is calibrated mainly with liquid developed market data not pose a problem to its direct usage in an emerging market setting? Emerging-market countries share some similarities to developed markets and their patterns. Often, their differences create unique opportunities and risks (Al Janabi 2006), which this paper explores through the Wilkens and Predescu (2016) model.

Since transparency and traceability are essential requirements for prudent risk measurement and management in especially emerging markets, the Wilkens and Predescu (2016) modelling approach was adopted and exclusively considered for estimating credit default risk in this paper.

Correlation Assumption

According to the regulatory requirements (briefly outlined in Figure 2) either credit spreads or equity spreads can be used to model default factor correlations in the IMA DRC. A credit default swap is a derivative product which serves as some form of insurance against the default of an underlying borrower or debt instrument. Ideally, credit default swaps are to be used to estimate the credit default correlations since it is arguably one of the most effective market indicators of credit default risk (Rikhotso and Simo-Kengne 2022). Furthermore, Lovreta and Pascual (2020) make the claim that credit default swap spreads reflect changes in credit risk more accurately and timeously than corporate bond yield spreads.

Unfortunately, corporate credit default swap data in the South African emerging market of interest are sparse (Clift 2020) and therefore do not always cover the general 2007–2008 credit market stress period. As a next resort, the fundamental asset value of an issuer company is usually considered a candidate for credit default correlation estimation. However, using the asset value as a proxy for credit default can be problematic, especially if the issuing company is a private company with no asset valuations in the public domain or is very young such that no historical asset information exists covering the 2008 crisis period. The last resort is to consider exchanged listed equity market data. Equity market data are relative to fundamental asset data, not sparse, easily accessible, cover a long history (especially for companies that have been around before 2008) and could be considered a

proxy to the asset prices and, finally, the credit default swaps. Unfortunately, the literature suggests that market equity returns can result in misstated asset returns (see, e.g., Düllmann et al. 2008; Laurent et al. 2016). However, using directly observed equity prices (compared to illiquid or non-existent corporate credit default spread data) to model default correlations could allow more transparency in understanding and risk managing the magnitude of the movements between obligors.

The problem of the lack of emerging market corporate credit default swap data (Rikhotso and Simo-Kengne 2022) motivates the exclusive use of equity data for default correlations in emerging markets. Even though the use of equity data is arguably a more crude approximation to the credit default dynamics, the choice of using equity data for emerging market default correlation at least allows compliance with the regulations. The following section describes a modelling framework that uses equity prices to proxy the emerging market default correlations.

*4.2. Framework*

Following the modelling framework of Wilkens and Predescu (2016), establish the response of the emerging corporate market on the global market. This is done by regressing the standardised[9] emerging market equity returns $r_{EM,t}$ on the global equity returns $r_{G,t}$:

$$r_{EM,t} = \beta_{EM} r_{G,t} + \epsilon_{EM,t} \tag{1}$$

Then, next establish the response of the various obligor industries on the global market. This is done by regressing the standardized emerging market equity industry returns $r_{In,t}$ on the global equity returns $r_{G,t}$. In other words,

$$r_{In,t} = \beta_{In} r_{G,t} + \epsilon_{In,t} \tag{2}$$

Using the betas estimated in Equations (1) and (2) determine the time series of the country and industry—specific returns (or residuals in Equations (1) and (2)). The sensitivity of each obligor j to the global return $\gamma_G$, systemic country factor $\gamma_C$ and industrial factor $\gamma_{In}$ are then estimated by the following regression equation,

$$r_{j,t} = \gamma_G r_{G,t} + \gamma_C \epsilon_{EM,t} + \gamma_{In} \epsilon_{In,t} + \epsilon_{j,t} \tag{3}$$

In fact, Equation (3) denotes the joint systemic factor return of obligor j. The reaction of obligor j to the joint systemic return, $\beta_i$ are then estimated using the following regression,

$$r_{j,t} = \beta_j (\hat{\gamma}_{G,j} \epsilon_{G,t} + \hat{\gamma}_{EM,j} \epsilon_{EM,t} + \hat{\gamma}_{In,j} \epsilon_{In,t}) + \epsilon_{j,t} \tag{4}$$

The quality of the systemic factor return regression per issuer (Equation (4)) is denoted by the correlation, or coefficient of determination $R_j^2$. The correlations are empirically derived and discussed in Section 5.1.

The obligor asset return can then be simulated using the following multifactor Gaussian copula (Merton 1974; Xiao 2009; Slime 2022):

$$\hat{r}_j = \text{sign}(\beta_j) \sqrt{\frac{R_j^2}{\Psi_j}} (\hat{\gamma}_{G,j} \hat{\sigma}_G Z_G + \hat{\gamma}_{EM,j} \hat{\sigma}_{EM} Z_{EM} + \hat{\gamma}_{In,j} \hat{\sigma}_{In} Z_{In}) + \sqrt{1 - R_j^2} \epsilon_j \tag{5}$$

with normalisation: $\Psi_j = (\hat{\gamma}_{G,j} \hat{\sigma}_G)^2 + (\hat{\gamma}_{EM,j} \hat{\sigma}_{EM})^2 + (\hat{\gamma}_{In,j} \hat{\sigma}_{In})^2$.

Here the standard deviation of the residuals in Equations (1) and (2) is given by $\hat{\sigma}_{EM}$ and $\hat{\sigma}_{In}$, whereas the standard deviation of the global factor is given by $\hat{\sigma}_G$. The $Z_G$, $Z_{EM}$, $Z_{In}$ and $\epsilon_j$ are all drawings from standard independent normal distributions.

As in the case of the Vasicek (2002) model, a jump to default is recorded when the estimated obligor return $\hat{r}_j$ (or default index as denoted in Figure 3) is lower than the inverse normal cumulative function of the default probability,

$$\hat{r}_j \leq N^{-1}(PD_i) \tag{6}$$

The loss distribution under the jump to default is therefore determined using the sum product:

$$L = \sum EAD_j \, LGD_j 1_{\hat{r}_j \leq N^{-1}(PD_j)} \tag{7}$$

Here the exposure at default is given by the EAD, the loss given default by the LGD, and 1 is the default indicator function signalling 1 when the one-year forward simulation resulted in default, and 0 otherwise.

## 5. Results and Discussion

### 5.1. Calibration

The regulator requires that the DRC model be updated at least weekly (see Figure 2). If the DRC is required to be updated weekly, then in order to be consistent, the default correlation must be calibrated using at least weekly equity data. As a result, the non-overlapping[10] weekly equity returns were determined with data from September 2007 covering the subprime crisis period. The correlation structure of the South African market to that of the global market (Equation (1) in Section 4.2) was calculated using the weekly return data. The correlation structure using the monthly equity returns was also determined for comparison purposes. The FTSE/JSE Top 40 equity index was used as a proxy for the South African market. A combination of the S&P500, FTSE100 and Euro50 indices was used as a proxy for the global market. The choice of the equity indices in the global proxy was motivated by Wilkens and Predescu (2016).

It was interesting to find that the regression coefficients suggest that the sensitivity of the emerging market to the global market ($\beta_{SA}$) and the quality of this correlation ($R^2$) are invariant under the sampled weekly and monthly returns. See the second column of Table 1. A good calibration is generally expected to be stable and not be required to be performed daily or whenever the DRC is calculated. Fortunately, Table 1 suggests that the modeller will be in precisely the same position using weekly or monthly data.

**Table 1.** Correlation structure of the country factor using weekly and monthly equity returns. The *t*-Stat is quoted at 95% confidence level.

| Calibration Data Frequency | $\beta_{SA}$ | *t*-Stat | $R^2$ |
|---|---|---|---|
| Weekly | 76.08% | 14.64 | 57.89% |
| Monthly | 75.57% | 6.925 | 57.12% |

Source: Own calculations using data from Thomson Reuters.

Another important facet of Table 1 is the reasonability of the correlations between the emerging and global markets. The size of for example the monthly $R^2$ (57%) can be argued to be reasonable based on the following argument. Wilkens and Predescu (2016) showed that the global factor does play a dominant role in developed countries. By studying a sample of 12 global markets, Wilkens and Predescu (2016) found that the likelihood of the developed markets to explain the US market was reflected in the high (65–90%) $R^2$ values. On the other hand, emerging markets do not necessarily fully explain the global factor. In fact, a sample of three emerging markets (Brazil, China and India) was found to represent (6%, 19%, and 9%), on average, only 11% of the US market (Pereiro 2010). Thus, in principle, a very strong correlation should not be expected between an emerging stock market and that of a developed global (US) market. Hence, for all practical purposes, the 57% correlation in Table 1 appears reasonable from an emerging market perspective.

In this paper, the calibration results are therefore determined using the monthly equity data with the reasonable 57% $R^2$ value obtained by covering the 2008/09 market stress period.

To numerically illustrate the calibration, five obligors were chosen, spanning the primary, secondary and tertiary economic sectors in South Africa. The obligors were purposefully sampled as each obligor was part of the top investment grade companies and had at least ten years of equity data publicly available. The calibration results are biased toward exchange-traded corporate issuers since, in South Africa, state-owned enterprises and municipalities do not have quality equity data in the public domain.

The estimated sensitivity of the obligor returns to the global, country and industry factors ($\hat{\gamma}_G$, $\hat{\gamma}_C$, $\hat{\gamma}_{In}$ in Equation (4) of Section 4.2) were then determined. Table 2 shows the sensitivities, as measured by the regression coefficients and their *t*-Stats (in italics). The coefficients are found to all be statistically significant at the 95% confidence level (*t*-Stat > 2) except for the country factor specific to the secondary economic sector obligors. The insignificant sensitivity of the secondary economic sector obligors to the country-specific factor does not deem the default correlation model (and its estimates) spurious. The estimated default model correlations are indeed plausible given the South African market is traditionally more of a primary- economic- sector-heavy market (Pereiro 2010; Wikipedia 2022) and is, therefore, not too sensitive to the secondary sector.

**Table 2.** Correlation calibration parameters and their *t*-Stats (in italics). The standard deviation of the industry specific factor is given by $\sigma_{In}$ whereas the standard deviation of the global factor $\sigma_G$ and the country specific factor $\sigma_I$ (see Equation (5) in Section 4.2) is set to 5.98% and 65.48%, respectively.

| Sector | Obligor | Global ($\gamma_G$) | Country ($\gamma_C$) | Industry ($\gamma_{In}$) | Industry Dev ($\sigma_{In}$) | β | $R^2$ |
|---|---|---|---|---|---|---|---|
| Secondary | 1 | 80.33% | −7.72% | 60.68% | | 83.47% | 52.00% |
| | | *3.4* | *−0.38* | *5.20* | 55.74% | *6.5* | |
| | 2 | 48.36% | −13.23% | 69.70% | | 117.11% | 49.52% |
| | | *2.03* | *−0.65* | *5.93* | | *5.86* | |
| Tertiary | 3 | 108.47% | −21.90% | 52.42% | | 64.40% | 54.62% |
| | | *11.45* | *−2.12* | *7.89* | 71.51% | *6.49* | |
| | 4 | 85.20% | −24.77% | 62.06% | | 86.01% | 68.17% |
| | | *6.80* | *−1.81* | *7.05* | | *6.50* | |
| Primary | 5 | 40.84% | 49.59% | 61.61% | | 93.77% | 72.92% |
| | | *1.54* | *1.50* | *7.87* | 81.60% | *9.71* | |

Source: Own calculations using data from Thomson Reuters.

The beta of the regression with the joint systemic factor returns (Equation (4) in Section 4.2) is all statistically significant. See column seven in Table 2. This means that the returns for the sampled emerging market obligors can be confidently proxied jointly by the three factors (global, country and industry) in the regression setting of Equation (4).

Interestingly, the industry dev ($\sigma_{In}$) increases as we move through the secondary and tertiary to the secondary sectors. The industry deviation is the standard deviation of the residuals in Equation (2) in Section 4.2. The industry deviation, therefore, reflects the distribution of the calibration model errors per obligor industry. The widening error distribution is expected, in the light of Munetsi and Brijlal's (2021) dispersion analyses showing that during a financial crisis the emerging market volatility (or width of the normal distribution) does increase as one move through the secondary and tertiary to the primary sectors.

Furthermore, the $R^2$ (last column in Table 2) is also increasing as we move through the secondary, tertiary, and primary sectors. The sampled emerging market is fundamentally more biased towards the primary and tertiary sectors (see Section 3). The primary and ter-

tiary sectors explaining most of the emerging market changes (via the higher $R^2$ in Table 2) can therefore be attributed to the unique nature of the emerging South African market.

Overall, the emerging market obligor default correlations, empirically calibrated under the modelling framework of Section 4.2, appear sound.

### 5.2. Experimental Testing

Before the emerging market default charge simulation model can effectively be used, some simulation testing is required. This section is therefore devoted to providing the simulation setup and the limiting correlation test cases that can be used to confirm the reliability of the emerging market DRC model.

### 5.2.1. Set-Up

The factor correlations defined in Table 2 were used as a base in the experimental simulation tests. It is well documented (Ferreiro 2016; Laurent et al. 2016; Wilkens and Predescu 2016; Slime 2022) that the extremely high 99.9% confidence level of the DRC requires many simulation paths to ensure reasonable accuracy. Therefore, as a start to testing the simulator, the minimum DRC number of simulations for good convergence was first established. To this end, the DRC per exposure per cent for an arbitrary portfolio was recorded after incrementally increasing the number of simulation paths. After approximately 100k simulation paths, a reasonable estimation uncertainty as measured by the width of the confidence interval of around 12% was noticed. See Figure 4. The uncertainty decreases to about 2% when 1 million simulation paths are used. The improved simulation convergence can also be ascribed to the preallocation of the random numbers in the program environment that allows optimisation via vectorisations and other matrix algebra (see Figure 3). For all practical purposes, 1 million simulation paths were used in all the DRC estimations.

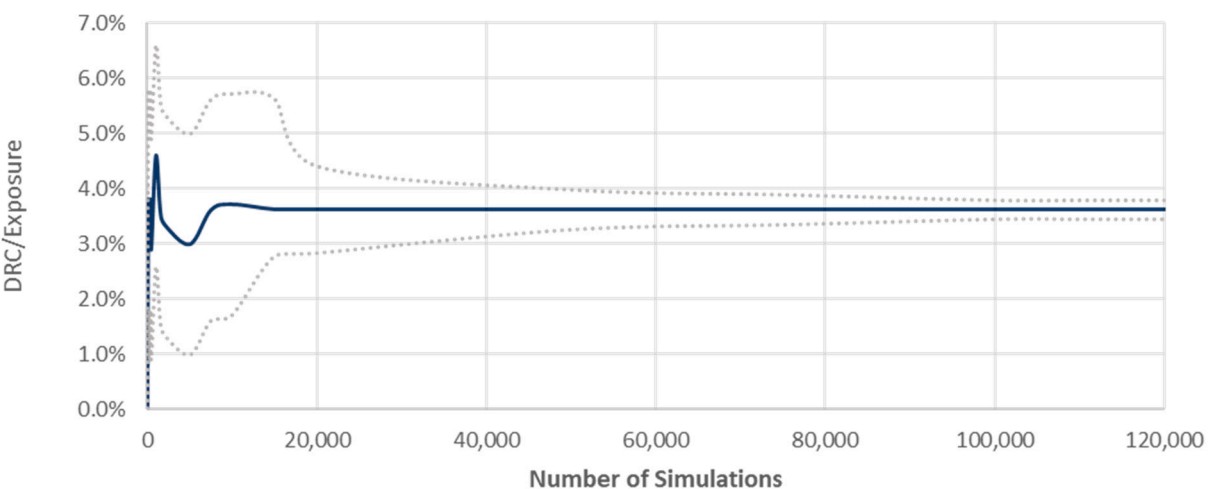

**Figure 4.** DRC sensitivity to the number of simulation paths. The estimated 0.1th percentile DRC (solid graph) as a function of the number of Monte Carlo simulation paths, together with the corresponding 95% confidence interval (dotted graphs).

Any model is merely just an abstraction of reality. It is, therefore, essential to take note of the assumptions underlying the model that can impose model risk (Hull 2015; Ferreiro 2016). In the model's derivation, the default index was assumed to evolve using random drawings from a Gaussian distribution (see Equation (5) in Section 4.2). The Gaussian copula is therefore an input to the emerging market default correlation simulation model (see Table 3). In general, bank portfolios are not normally distributed (Hull 2015; Ferreiro 2016; Wilkens and Predescu 2016). Therefore, the accuracy of the results may need to

be further tested using other underlying distributions, such as the Student or Gumbel distributions (Slime 2022). Another assumption underlying the simulation is the use of the Mersenne Twister pseudo-random number generator. However, since most modern simulation packages uses this random number generator the default simulation model risk the Mersenne Twister poses to the precision of the DRC is relatively low (Hull 2015; Ferreiro 2016).

**Table 3.** Parameters used in the default simulations.

| Random Simulation | |
| --- | --- |
| Paths: | 1 million |
| Underlying distribution: | Gaussian |
| * Algorithm: | Mersenne Twister |
| Seed: | 456,789 |

* Random number generator algorithm.

### 5.2.2. Design

All the tests were performed by considering five different obligors, each having issued three instruments (representing credit-bearing instruments such as corporate bonds of various maturities). To infer confidence in the emerging market default simulation model, four test cases were performed. The four cases were divided into two market scenarios. The one "Market Crash" scenario entails a relatively high correlation between issuers, as seen during the 2008–2009 global financial crisis or the COVID-19 global epidemic. The other "Fully Diversified" market scenario entails a lower correlation, common to a normal or non-extreme market.

### Market Crash Scenario

The two of the four cases designed to mimic an event analogous to a market crash have defaults that are totally deterministic, and consequently all the market betas converges into one. In this scenario, the $R_j^2$ and all the factor sensitivity parameters in Equation (5) of Section 4.2 are therefore set to unity. In this scenario, the obligor asset return Equation (5), therefore deterministically reduces to:

$$\hat{r}_j = \frac{1}{\sqrt{3}}(Z_G + Z_{SA} + Z_I)$$

such that the loss distribution (Equation (7)) becomes:

$$L_j = n \times EAD \times LGD \times 1_{\frac{1}{\sqrt{3}}(Z_G + Z_{SA} + Z_I) < N^{-1}(PD_i)} \tag{8}$$

Here n is set to be 15 (5 obligors $\times$ 3 instruments). The EAD is set to 100 monetary units whereas the PD and LGD is defined separately for each of the Market Crash Scenario test cases.

### Fully Diversified Scenario

The other two of the four cases are designed to mimic an event where the portfolio becomes fully diversified. In these test cases, there are simultaneous defaults of numerous dependent issuers (Laurent et al. 2016; Slime 2022) and, hence, no dependence on the systemic factors. As a consequence, the correlation $R_i^2$ and all the factor sensitivity parameters in Equation (5) of Section 4.2 become nil. In the fully diversified scenario, the obligor asset returns, and Equation (5), thus, reduces to:

$$\hat{r}_j = \epsilon_j$$



Consequently, the loss distribution (Equation (7)) becomes:

$$L_j = n \times EAD \times LGD \times 1_{\epsilon_j < N^{-1}(PD_j)} \tag{9}$$

Here n is set to be 15 (5 obligors $\times$ 3 instruments). The EAD is set to 100 monetary units whereas the PD and LGD are defined separately for each of the Fully Diversified Scenario test cases.

### 5.2.3. Test Results

The default risk charge is not only used for regulatory purposes. Klaassen and van Eeghen (2009) claim that at different confidence intervals, the default risk charge can also approximate the credit add-on in an economic capital setting. From a trading book perspective, the variability in the DRC at varied confidence levels is interesting, especially because Laurent et al. (2016) showed that for diversified and hedge portfolios the DRC almost doubles when the confidence level is moved from 99% to 99.9%. For purposes of gauging the soundness and usefulness of the emerging market simulation model, the default risk charge at various percentiles is given in Table 4.

**Table 4.** Simulated default risk charges at various distribution percentiles. The simulations were done by adjusting the correlation parameters to coincide with that of a "Market Crash" and that of the "Fully Diversified" portfolio. The regulatory prescribed 99.9% DRC is given in grey.

| Scenario: | Market Crash | | Fully Diversified | |
|---|---|---|---|---|
| Correlation: | Perfect | | Zero | |
| PD: | 100% | 100% | 0 | 50% |
| Percentiles | DRC | DRC * | DRC | DRC |
| 100 | 1500 | 900 | 0 | 1300 |
| 99.9 | 1500 | 900 | 0 | 1000 |
| 99.8 | 1500 | 900 | 0 | 1000 |
| 99.5 | 1500 | 900 | 0 | 900 |
| 98 | 1500 | 900 | 0 | 800 |
| 97 | 1500 | 900 | 0 | 800 |
| 96 | 1500 | 900 | 0 | 800 |
| 95 | 1500 | 900 | 0 | 800 |

* For this test case the LGD for only the first two obligors are set to nil. The remaining obligors has LGD set to 100%.

In the first "Market Crash" scenario test case (column two in Table 4), the loss function (Equation (8)) becomes (5 obligors $\times$ 3 instruments $\times$ 100 EAD $\times$ 100% LGD = 1500 monetary units) since all the obligors jump to default. Furthermore, the loss distribution is therefore flat, such that the DRC is 1500 monetary units at all percentiles. Thus, as expected, in this case where the market crashes, and the default probability and the LGD is set to 100%, the simulation correctly results in 1500 monetary units of default risk charge.

In the second "Market Crash" scenario test case (column three in Table 4), the loss function is the same as the previous case, except that the first two obligors have LGD set to nil. The loss function, therefore, becomes (3 obligors $\times$ 3 instruments $\times$ 100 EAD $\times$ 100% LGD) 900 monetary units of default risk charge, as expected.

In the first "Fully Diversified" scenario test case (column four in Table 4), the default probability is set to nothing, such that no charge is being raised, and the DRC is nil at all percentiles, as expected.

In the second "Fully Diversified" final scenario test case (column five in Table 4), the default probability is now set to 50%. The loss function is now fully stochastic, so

the default index is only the second term in Equation (5). The simulation distributional properties are, therefore, important in this test case. As expected, the simulation default risk charge decreases as the percentile decreases, similar to that found by Wilkens and Predescu (2016).

For all practical purposes, the limiting test cases presented above show that the simulation-based DRC appear reliable for estimating the emerging market default risk charge.

## 6. Conclusions

A default risk charge model for emerging markets that follows the regulatory guidelines was presented. Emerging markets generally do not have good-quality data. As a result, underlying data assumptions and approximations are used in credit risk estimation models. One assumption that is made in this research is the fact that in an emerging market, the equity returns are a reasonable approximation to the credit default market dynamics. In this paper, equity data were exclusively used to calibrate the default correlation for the emerging South African market. A transparent yet mathematically eloquent default correlation modelling framework, similar to that in Wilkens and Predescu (2016) was derived using the emerging market equity data. A purposeful sample of obligors was used to empirically perform the default calibration. The results showed that the derived emerging market default calibration is sensible, except for the secondary economic sector. Limiting market test cases were performed to confirm the reliability of the simulation-based DRC under the default calibration.

The key insights of this paper are as follows:

-   The emerging market data, generally of poor quality and not abundant, prompted using equity returns to derive the default correlations. Even though this data choice seems plausible from an emerging market perspective, it may introduce some DRC variability. Using an arbitrary data choice (equity returns or CDS spreads) regarding the default calibration could lead to ill-favored variability in the DRC (Laurent et al. 2016). Apart from having two IMA DRC approaches, one for emerging markets and one for developed markets—articulating what type of data to be used, another response could be to require all banks to disclose the choice of data used for the default calibrations.
-   From a policy perspective, the regulator may consider dropping the 0.03% default probability floor as it may be too punitive for emerging markets. The impact of dropping the 0.03% floor is likely to be material for banking portfolios that tend to comprise significant positions in securities of well-rated sovereigns and corporates (Wilkens and Predescu 2016). In fact, in some emerging markets (like South Africa) banking portfolio default calibration data is not always available due to a scant or sporadic history of corporate credit defaults. The lower the likelihood of default, the more observations are required to produce reliable default estimates (Ferreiro 2016). Therefore, the data problem in emerging markets could translate into a default model risk problem. The probability floor of 0.03% results in a shorter distance to default for obligors with a much lower default probability (according to Equation (6) in Section 4.2). The probability floor could be thought of as some conservatism introduced to alleviate the data reliability problem. Unfortunately, the probability floor may also be argued to be introducing some fictitious conservatism in the default charge, especially for emerging markets with very well-rated bank portfolio obligors (having a default probability much less than 0.03%).
-   For emerging markets, the regulator may consider allowing the use of standardised LGDs (BCBS 2016) (25% for covered bonds, 75% for senior bonds, and 100% for non-senior instruments including equity) in the IMA DRC. Using the standardised LGDs in an emerging market DRC model could result in some margin of conservativism. The conservatism introduced in the emerging market DRC model is preferable from a risk perspective compared to the extra possible data uncertainty injected into the DRC via the modelling and estimating of the systemic factor-dependent LGDs. Moreover, the conservatism introduced by using the standardised LGDs in the emerging market

IMA DRC model is deterministic. Thus, considering the use of standardised LGDs in the emerging market DRC model could move the regulator, supervisors, and banks closer to getting emerging market institutions to comply better and confidently adopt the IMA DRC.

The limitations and new avenues for research are presented as follows:

- Data from only one emerging market were used to calibrate the default correlations. A possible avenue for future research is to perform the default calibration on multiple emerging markets. This will not only allow direct comparisons of the calibration parameters but will also give a good idea of the measurement and management of the stability of the calibration parameters over several emerging markets.
- Another limitation of the research is that the correlation calibration excluded data covering the COVID-19 epidemic period. Another possible avenue for future research is to include the epidemic period in the ten-year calibration data period. Including the epidemic data period in the calibration will allow the direct comparison of the default charge estimate under the 2008–2009 crisis and the more current epidemic period. Such a comparison can be helpful from a market risk stress management perspective.
- The regulations suggest that the LGDs used in the DRC should ideally depend on the systemic factors used in the default calibration. See Figure 2. However, this research was limited to using fixed LGDs in the DRC simulation, which is not based on the derived systemic factors. The fixed LGDs were used in the emerging market DRC setting to allow for a reasonably manageable set of data assumptions in the model. Future work on emerging-market DRC modelling could explore stochastic or systemic factor based LGDs similar to the framework outlined in Section 4.2. This exercise will probably facilitate an academic contribution in that, given the data problems common to emerging markets, it could quantify and empirically show the degree to which an emerging market DRC becomes inaccurate due to too many data assumptions and approximations.

**Supplementary Materials:** The following supporting information can be downloaded at: https://www.mdpi.com/article/10.3390/jrfm16030194/s1.

**Funding:** This research received no external funding, and the APC was funded by the University of South Africa.

**Data Availability Statement:** Data used for this research are available as Supplementary Material.

**Acknowledgments:** Thanks to Antonie Kotzè, an affiliate to the Australian Institute of Physics, for providing thought leadership in quantitatively modelling the regulatory rules. Many thanks to Badreddine Slime from the University of Technology of Compiègne in France for fruitful discussions on distilling the regulations. Thanks to Kenneth Koh from Cornell University in New York for help with understanding the operational data requirements. Thanks to Geoffrey Kimetto from INSEAD in France for helpful comments in the initial draft of the paper. Thanks to an anonymous referee for helpful suggestions on the final structure of this paper. The views expressed in this paper are solely the responsibility of the author and should not be interpreted as reflecting the views of any of the related persons or their institutions.

**Conflicts of Interest:** The author declares no conflict of interest.

## Appendix A

Consider a credit exposed portfolio with K positions. The portfolio loss (given a jump to default) is given by:

$$L(Z, \varepsilon) = \sum_{k=1}^{K} EAD_k \, LGD_k I_{X_k \leq \varphi^{-1}(p_k)}$$

Here $I_{X_k \leq \varphi^{-1}(p_k)}$ is the default binary indicator function with simulated default index (Laurent et al. 2016):

$$X_k = \beta_k Z + \sqrt{1_d - \beta_k{}^T \beta_k} \ \varepsilon_k$$

where the systemic factor $Z \sim N(0, \Sigma_Z)$.

$$X_k \in \Re^{Kx1}, \ \beta_k \in \Re^{KxJ}, \ \varepsilon_k \in \Re^{Kx1} \text{ and } 1_d$$

is the identity matrix, with J the number of factors.

The prescribed "two factors" can be introduced by doing an eigen-value decomposition on the correlation matrix $\Sigma_Z$.

$\beta_k$ is the factor loading matrix either from a factor (Laurent et al. 2016) or regression (Wilkens and Predescu 2016) analysis. The random drawing $\varepsilon_k \sim N(0, 1)$.

The default threshold is $\varphi^{-1}(p_k)$ is designed such that $p_k$ is the marginal default probability of obligor k, and $\varphi$ is the standard normal cumulative function.

The default risk charge of a portfolio with different obligors is then defined as the 99.9 percentile loss given the jump(s) to credit default. Moreover, the DRC is a VaR type calculation based on the average losses simulated over a 1 year forward horizon:

$$DRC = VaR_{99.9\%} [L(Z, \varepsilon)]$$

## Notes

[1] Default correlation can be thought of as a measure of how in sync issuers of debt default in the event of a global, country or industry wide crisis, like the subprime credit crisis experienced in the years 2008 to 2009.

[2] Accounting for the default charge by adding on to the overall risks are permissible since the default charge is comparable to a VaR metric (Rodrigues and Maialy 2018).

[3] This limitation arises because credit correlations in markets are generally driven by more than one economic factor.

[4] Interestingly, the introduction of the credit risk measure that excludes migration came against the backdrop of the global 2008–2009 financial crisis showing that most losses to the banking sector were caused not by default but by credit migration (Prorokowski and Prorokowski 2014).

[5] Spread risk is more concerned with default expectations, whereas the default risk charge is concerned with the capital at risk in the event of a jump-to-default. The jump-to-default risk is analogous to jumping from a cliff (compared to a slow credit migration that allows time for credit spreads to reprice and losses to be cushioned).

[6] Banks are required to seek formal approval from the supervisor to use the internal models approach for each desk; otherwise non-qualifying desks will be subject to the standardised capital framework.

[7] It is counterintuitive that a default risk measure was introduced to supplement a market risk measure in the traditional sense. However, modern portfolios generally contain market and credit risks that are not fully explained by the original market capital framework (Ferreiro 2016).

[8] When the correlation between issuers and the concentration risk is higher, one generally expects the DRC to increase (Slime 2018). This proportionality is not fully preserved in the standardized DRC.

[9] Time series standardised to reflect a mean of 0 and standard deviation of 1.

[10] Overlapping return correlations can be prone to autocorrelation effects. Moreover, the use of overlapping return correlations can distort the correlations to such an extent that the maximum, true peak correlations can go undetected.

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
