# Peer review of "Emerging Market Default Risk Charge Model"

_jrfm, doi:10.3390/jrfm16030194_

Round 1
Reviewer 1 Report
(1) Authors should provide the economic significance of the results to highlight their relevance.
(2) Qualitative econometric methodology is unclear.
(3) The empirical methodology adopted in this article is not clear. The authors should explain more the empirical work and how they obtained the results.
(4) What are the main research implications from the perspectives of investors and financial market authorities?
(5) I think that the literature mobilized in the section relating to the hypotheses should be presented in a more developed way.
(6) At the conclusion level, it is important to present the limits and as well as the new avenues for future research.
(7) Authors should provide more details on the practical/policy implications of their results.

Author Response
Point 1: Authors should provide the economic significance of the results to highlight their relevance.
Economic significance of results carefully incorporated. See e.g. section 5.1
Point 2: Qualitative econometric methodology is unclear.
Calibration method is explained mathematically in section 4.2. The simulation process is demonstrated in two ways. Schematically in Figure 2 and algebraically in Appendix A.
Point 3: The methodology adopted in this article is not clear. The authors should explain more the empirical work and how they obtained the results. Noted. The results section was expanded with this comment in mind. See e.g. Table 3.
Point 4: What are the main research implications from the perspectives of investors and financial market authorities? This has been carefully articulated in section 6.
Point 5: I think that the literature mobilized in the section relating to the hypotheses should be presented in a more developed way. Agree, for this reason sections such as 2.2.1 was introduced.
Point 6: At the conclusion level, it is important to present the limits and as well as the new avenues for future research. The limitations and new avenues for future research were incorporated in section 6.
Point 7: Authors should provide more details on the practical/policy implications of their results. This was also done in section 6.

Reviewer 2 Report
Overall the paper is interesting, but a number of elements need to be corrected, which I describe below:
1. There is a need to better explain the relevance of simulation models, to explain the effect that poor quality can have on the quality of the simulation models, and to explain how they can have a positive effect on the quality of the simulation models.
In that sense, better explain the VaR-->IRC-->DRC-->IMA DRC step.
2. The model talks about 99.9%, and I understand that this parameter is used as described in the literature. Explain that it is a parameter, that it can have another value and explain why such a high confidence level is used, why not use 99% or 95%?
3. When talking about correlations between assets being an important topic in the literature, mention what happens in the case of extreme events, exemplified by concrete cases such as the Covid crisis or the Russia-Ukraine war.
4. In the simulation section it is necessary to place a table with the parameters used in the simulation.
5. What happens in simulated high, medium and low correlation scenarios? An additional round of experimentation is needed considering different asset correlation scenarios. Or a better explanation of the experimental scenario, which is a bit messy.
6. On table 1 it is mentioned that an R2 of 57% is good. There is a lack of arguments to say this, be it statistical criteria, literature support, etc.
7. Figure 4 should be improved, either by showing a smaller number of simulations or the scale in logs, as after the 40000 iteration it does not provide more information.
8. It should be mentioned in the text what is meant by emerging countries, is it based on a scale, a criterion of some organization, is there a list of countries beyond South Africa? what about Latin America, Asia?
9. The results need to be checked against the existing literature, so the real contribution of the paper is not clear. This should be described in the introduction, discussion and conclusions of the paper.
10. Substantial improvement of the literature cited is required.

Round 2
Reviewer 2 Report
They have responded satisfactorily to the comments